# Remote Sensing Image-Change Detection with Pre-Generation of Depthwise-Separable Change-Salient Maps

**Bin Li [1], Guanghui Wang [1,2,*], Tao Zhang [2], Huachao Yang [1] and Shubi Zhang [1]**

1. School of Environmental and Spatial Informatics, China University of Mining and Technology, Xuzhou 221116, China
2. Land Satellite Remote Sensing Application Center, Ministry of Natural Resources of China, Beijing 100048, China
* Correspondence: wanggh@lasac.cn

**Abstract:** Remote sensing change detection (CD) identifies changes in each pixel of certain classes of interest from a set of aligned image pairs. It is challenging to accurately identify natural changes in feature categories due to unstructured and temporal changes. This research proposed an effective bi-temporal remote sensing CD comprising an encoder that could extract multiscale features, a decoder that focused on semantic alignment between temporal features, and a classification head. In the decoder, we constructed a new convolutional attention structure based on pre-generation of depthwise-separable change-salient maps (PDACN) that could reduce the attention of the network on unchanged regions and thus reduce the potential pseudo-variation in the data sources caused by semantic differences in illumination and subtle alignment differences. To demonstrate the effectiveness of the PDA attention structure, we designed a lightweight network structure for encoders under both convolution-based and transformer architectures. The experiments were conducted on a single-building CD dataset (LEVIR-CD) and a more complex multivariate change type dataset (SYSU-CD). The results showed that our PDA attention structure generated more discriminative change variance information while the entire network model obtained the best performance results with the same level of network model parameters in the transformer architecture. For LEVIR-CD, we achieved an intersection over union (IoU) of 0.8492 and an F1 score of 0.9185. For SYSU-CD, we obtained an IoU of 0.7028 and an F1 score of 0.8255. The experimental results showed that the method proposed in this paper was superior to some current state-of-the-art CD methods.

**Keywords:** change detection; attention mechanism; deep learning; feature enhancement

## 1. Introduction

Remote sensing change detection (CD) is the process of identifying changes by observing the state of an object or phenomenon at different times [1]. The definition of change in this process varies depending on the "target of interest" and can usually be classified into binary and multiple CDs [2] such as building CD [3], mudflow and landslide CD [4], building damage assessment [5,6], land cover CD [7], and deforestation [8,9]. The development of satellite imagery technology has facilitated the collection of massive amounts of remote sensing data with high spectral, spatial, and temporal resolutions. However, the increase in image resolution and the addition of various forms of change due to different environmental conditions in remote sensing images makes CD a challenging task [10]. Therefore, the key to CD methods is to reduce interference from extraneous changes in increasingly complex environments. Early traditional CD methods typically compared pixel values by using machine learning design features [11]. Although some of them performed better with the setting of thresholds and manual design of features during the comparison process, their detection accuracy relied heavily on the quality of homogenous remote sensing images and was vulnerable to noise variations.

With the rapid development of deep learning technology in the field of image analysis, scholars have continuously proposed using deep learning technology to address the problem of CD in high-resolution remote sensing images [11,12]. Compared with CD methods based on simple mathematical models, deep learning models have more complex network structures and can automatically learn high-level semantic features from large amounts of big data. For bi-temporal, high-resolution remote sensing geospatial object CD, most of the current network change detectors are based on deep convolutional networks [13], especially fully convolutional Siamese networks (FC-Siam) [14]. FC-Siam uses weight-sharing encoders to extract depth features and then uses feature difference decoders to detect object changes from the perspective of the encoder–decoder architecture. Most different approaches are then modified in terms of how the different features extracted from the dual encoders are consumed (or compared) to produce a CD prediction layer. These approaches focus on three main aspects [15]: the encoder (i.e., the use of a pretrained deep network as an encoder) [16–18]; the decoder (i.e., a recurrent neural network (RNN) and temporal attention-based decoder) [17,19]; and the training strategy (i.e., deep supervision of multiple outputs) [20,21]. Here, we focus on the decoder part of the network, which is important to the network in terms of the generation of discriminative and robust change features. The decoder can be divided into two steps: (1) extracting features with unique change information (i.e., reinforcing features by constructing an attention mechanism); and (2) designing a decision function to generate a change map based on the extracted features (i.e., further mining the temporal dependencies between images by using an RNN to process the cascaded feature pairs using convolution). However, because it is affected by the scale size related to the images, the related design of the decoder often introduces additional performance overhead, causing difficulties in model building.

To address the above problems, we proposed a bi-temporal image CD network based on the pre-generation of depthwise-separable change-salient maps (PDACN). First, we proposed a dual encoder/single decoder scheme with a lightweight depth feature encoder to extract features of different scales and widths. A lightweight multiscale feature generation encoder structure was then constructed in which the model paid more attention to change information discrimination in the region of interest, thus reducing the effect of noise from more complex environmental information in high-resolution images. The network learned what to emphasize and how to extract features that described change at a higher level. Finally, we validated our proposed module by applying it to publicly available datasets with two different levels of change complexity.

In summary, this paper makes two main contributions.

(1)　A novel end-to-end CD method for remote sensing images is proposed.
(2)　A new convolutional attention structure dedicated to CD that enables the CD network to pay more attention to change information extraction from change regions while taking into account channel and spatial adaptability is proposed.

## 2. Related Work

Here, change-detection methods are broadly divided into two categories: traditional and artificial intelligence (AI)-based [11]. Traditional methods focus on the artificial design of the relevant features to obtain the change intensity information, while numerous remote sensing studies have shown that AI-based change-detection methods outperform traditional methods in terms of feature extraction [22,23]. Therefore, we focus here on the present development of AI techniques in the field of change detection, especially deep learning-based bi-temporal change detection. However, the generic change detection process is still of importance in deep learning-driven change detection. The key lies in the two main aspects of designing a decision function to generate a change map and extracting features with robust semantic information. In the following, we provide an overview of the related work in recent years around these two aspects.

## 2.1. Bi-temporal Change Detection Using Deep Learning

Existing CD benchmark datasets [24–28] have been constructed based on bi-temporal supervised learning, which requires change labels for remote sensing images of the same region over different periods. Network change detectors can be classified according to spatial units: networks based on the block structure and network structures for pixel-by-pixel classification [29].

Block-structure-based networks usually incorporate image classification tasks. Zhang et al. [23] proposed a CD method based on image blocks that divided images into 28 × 28 or other sizes of image blocks and then inputted the blocks into a convolutional neural network (CNN) composed of two identical structures sharing weights. The two feature vectors outputted from these two CNNs were fused to generate a feature vector covering the change information, and finally, two hidden layers were used to classify the change category of the central pixel. Ye et al. [30] also proposed a supervised CNN structure, AggregationNet, which took two remote sensing image blocks with different phases as inputs and outputted a two-dimensional vector by constructing a Siamese network at the feature fusion level. Finally, the class information of each of the two image blocks was outputted to obtain the classes of each to determine whether a change had occurred. Wiratama et al. [31] proposed a dense Siamese CNN architecture for a network consisting of two change detectors as independent convolutional subnetworks. The input was a 40 × 40 pixel image block and the output was the Euclidean distance metric of the two network outputs. The center pixel of the image block was classified as exhibiting no change when the value of the Euclidean distance metric approached 1, corresponding to a label of 0. Although the image-block-based approach transformed CD into a classification problem and differed from post-classification CD, it not only determined whether there was a change, but also directly obtained the type of change. However, block-structure-based methods are sliding-window approaches; they have very slow inference times and are inefficient because the same region is visited multiple times (there is a large overlap between image blocks adjacent to intermediate pixels).

To reduce the difficulty of detection, network structures for pixel-by-pixel classification usually consist of some end-to-end methods that directly classify each pixel on the feature map. Most of them utilize the classical fully convolutional network (FCN) [32] and U-Net [33]. CDNet [34] and fully convolutional–early fusion (FC-EF) use the hourglass codec structure of the FCN to predict regions of change. This type of approach uses two images detected directly as inputs to the network and is very susceptible to noise variations. FC-Siam-Conc [35] and FC-Siam-Diff [35] employ a Siamese network structure with a double decoder–single decoder. The Siamese network maps images to a specific feature space, which can restrict the noise in images input to the decoder that processes high-level semantic information, but fails to consider the temporal dependence between images. BiDateNet [36] uses long short-term memory (LSTM) to address the problem of inaccurate image registration. In addition, to further enhance the representation of the network model and make the network pay more attention to the change information of the region of interest, the spatiotemporal attention-based network (STANet) [17] obtains change maps via metric learning by using the spatiotemporal attention mechanism to obtain more discriminative features.

## 2.2. Designing a Decision Function to Generate a Change Graph

The overall process of generating change maps based on extracted features can be divided into two forms: metric-based learning and classification-based learning. The metric-based approach obtains a change map by measuring the distance between features. Since the distance between unchanging features is relatively short and the distance between changing features is relatively long, the metric-based approach can further distinguish between "changing" and "unchanging" features in the decision phase, thus mitigating the effect of pseudo-change and improving CD accuracy. STANet [17] obtains a change map by finding the Euclidean distance between different features and uses a spatiotemporal

attention mechanism to obtain more discriminative spatiotemporal features. DSAMNet [20] includes a metric module to learn change maps from the distances between pairs of bipartite temporal feature maps in a low-dimensional embedding space. CEECNet [37] introduces a new ensemble similarity metric that is a variant of the Dice coefficient, which can be steeper than the standard Tanimoto metric, thus providing a more fine-grained similarity metric between layers. However, the improvements that these methods can achieve are limited due to the limitations of time dependence in distinguishing pseudo-variations that are very confusing in appearance.

Classification-based methods tend to fuse twin features and then use a multilayer convolutional structure to reduce the number of channels for both variable and invariant types of channels; this tends to be the most commonly used feature-fusion method. The bi-temporal image transformer (BIT) [38] uses Transformer to construct a spatiotemporal attention mechanism and to fuse features by taking absolute values of the enhanced features. The feature constraint change detection network (FCCDN) [21] contains a deep dense fusion module (DFM) that consists of two branches: a sum branch and a difference branch. The sum branch is used to enhance edge information and the difference branch is used to identify change regions.

### 2.3. Extraction of Features with Robust Semantic Information

Extracting features with unique variation information and more discriminative features is important in mitigating pseudo-variation such as false alarms caused by differences in external factors; e.g., light and scale differences between dual-time inputs. Therefore, many attempts have been made to generate more discriminative features to overcome this problem. Peng et al. [39] developed UNet++ with dense skip connections, aiming to learn more effective features from multiscale semantic information. Since they have an excellent ability to capture temporal dependencies between diachronic images, RNNs have been used to acquire features with spatiotemporal information. Song et al. [40] proposed a recurrent 3D FCN for hyperspectral images that combined the advantages of a 3D FCN and an LSTM network. Papadomanolaki et al. [36] proposed BiDateNet to enhance the temporal information between diachronic images by integrating LSTM into the U-Net structure. The improvements that can be achieved with these methods are limited due to the limitations of temporal dependence in distinguishing pseudo-variations that are very confusing in appearance. Therefore, some attention-based mechanisms have been applied in this process. PGA-SiamNet [41] integrates a global collective attention mechanism into a pyramid FCN to capture architectural change. Chen et al. [42] used a dual attentional CNN with a VGG16 pre-trained network for the feature encoder. The attention module they used for vision incorporated both spatial and channel attention and was introduced by Vaswani et al. [43]. A dual-task constrained deep Siamese convolutional network (DTCD-SCN) [44] was designed as a dual-task constrained CD network that was responsible for building extraction and changing building extraction. To further improve the representation capabilities between features, a dual attention module (DAM) that takes full advantage of the interdependence between channel and spatial location has been proposed.

## 3. Methods

In this section, we first give a general introduction to the proposed model followed by a detailed description of each modular part of the model. Finally, a brief description of the optimization strategy of the model is given.

### 3.1. Overall Structure

The proposed network structure is shown in Figure 1. It had the form of a Siamese network structure and consisted of three parts: an encoder, a decoder, and a classification head. To learn representative features of images from different time periods, the weight-sharing feature extractor automatically extracted representative multiscale features from each of the two images. The decoder then processed these multiscale features from different

time periods to recover the image features while reconstructing the detailed parts of the image features. Notably, to eliminate possible semantic errors on the feature maps from different periods, we use pre-generated depthwise-separable change extraction (PDC) to generate a change-saliency map for the initial change location and then multiplied the saliency map by the original feature map to implement a common attention mechanism to extract more robust change features. Therefore, we named this structure the PDA decoder. The role of the classification head part was to reduce the number of feature map channels to two types (variable and invariant) using a multilayer convolutional structure and finally generate a change binary map via the argmax operation. In the case of multiple classifications, the output could be chosen as the number of channels with the same number of categories. Here, we called the entire network PDACN.

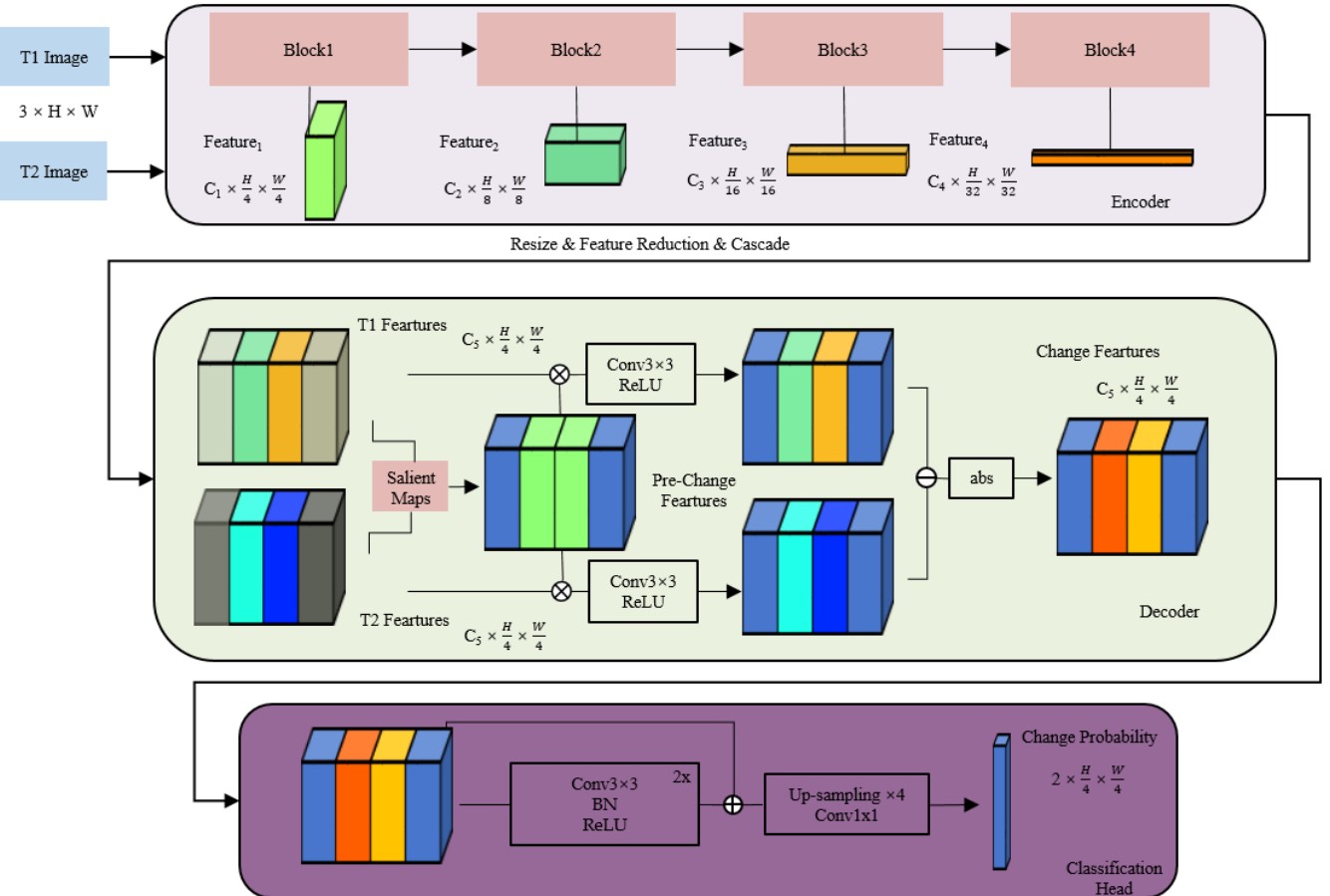

**Figure 1.** The overall structure for our change detection network.

Let $I_1$ and $I_2$ denote pairs of images of the same region at different points in time and let L denote labels with change annotations. The flowchart of PDACN can be summarized as follows.

(a)    First, $I_1$ and $I_2$ were inputted to the same encoder to obtain semantic features with different depths and resolutions $T_1$ (Feature$_1$, Feature$_2$, Feature$_3$, and Feature$_4$) and $T_2$ (Feature$_1$, Feature$_2$, Feature$_3$, and Feature$_4$).

(b)    Next, the features with the same timestamp were combined into one by $1 \times 1$ convolution to obtain $T_1$ features and $T_2$ features with the same dimension $C_5$ and the same width and height ($\frac{H}{4}$, $\frac{W}{4}$). To obtain more discriminative features, we first used the PDC module to initially locate the change regions to obtain pre-change-salient maps and then multiplied them by the bi-temporal features and passed them through a $3 \times 3$ convolution and a rectified linear unit (ReLU) function. After the above operation, we could eliminate the influence of unchanged regions on the final network

results and thus reduce the pseudo-variation. Note that we did not use batchnorm or layernorm operations in the further processing of the $T_1$ features and $T_2$ features.

(c) Finally, we used two-layer convolution, batch normalization, a convolution module built with the ReLU and a residual connection structure to complete the correction operation on the change feature map, which could further enhance the fitting ability of the model. After upsampling the corrected change features to the original remote sensing image size, the final change probability map could be obtained by using a $1 \times 1$ convolution. The change probability map could be directly compared with the label L to calculate the loss.

### 3.2. Encoder

Many research results have shown that using pre-trained deep networks as encoders facilitates model convergence [16,17]. Pre-trained network structures often benefit from some neural network structures in the non-remote sensing domain and can be generally classified into two architectural forms based on convolution and transformation. As depicted in the encoder section in Figure 1, both network structures had similar hierarchical feature-extraction structures and were capable of extracting representative features; the difference lay only in the mechanism of feature extraction.

As shown in Table 1, from the perspective of model capacity, we preferred the encoder structure with fewer parameters and a faster computation speed and constructed three encoding forms of network structures based on this. For the convolutional architecture, we selected ResNet18 as the standard encoder and made the appropriate modifications. Unlike the original ResNet18 [13] architecture, the initial convolutional step was set to 1. Therefore, the resolution of the feature map of the first block in the figure became 1/2 that of the original one, which led to a greater computational burden. As the number of blocks increased, the resolution of the feature map became 1/2 that of the previous level layer by layer and the number of channels became twice that of the previous level layer. For the transformer architecture, we adopted SegFormer-b0 [45] as the encoder and did not make any modifications. S3 indicates that we obtained only the feature maps of the first three block outputs for decoding, while S4 indicates that all the features of the block outputs were involved in decoding. Therefore, the final output feature channel numbers C1, C2, C3, and C4 of the ResNet18-S4 and SegFormer-b0 encoders were (64, 128, 256, 512) and (32, 64, 160, 256), respectively.

**Table 1.** Capacity analysis of the entire model with three different encoders.

| Encoder | Input Size | Params. (M) | FLOPS (GFLOPS) |
|---|---|---|---|
| ResNet18-S3 | (3, 1024, 1024) | 3.65 | 510.60 |
| ResNet18-S4 | (3, 1024, 1024) | 12.13 | 588.51 |
| SegFormer-b0 | (3, 1024, 1024) | 4.22 | 89.31 |

### 3.3. PDA Decoder

For a Siamese CD network architecture, constructing a robust bi-temporal feature fusion module to minimize pseudo-variation is the most critical part. Two problems need to be solved [21]: (1) different-period images often suffer from color inhomogeneities and spatial position shifts caused by alignment errors; and (2) the complex distribution of features in remote sensing images greatly increases the number of categories of change information in images in different periods. To address the problem of bias between such diachronic features, some researchers have constructed a series of change feature-extraction mechanisms based on the use of direct subtraction and tandem methods to fuse features. However, these methods are usually affected by the width and resolution of the features to some extent, which introduces a greater computational burden and results in more memory consumption.

Here, we propose a simple and effective PDA feature-fusion module. The purpose of PDA was to give more attention to the regions that changed in the feature space during

change-information extraction under the condition of ensuring spatial adaptation and channel adaptation and to pay less attention to the regions that did not change to reduce the effects of pseudo-change due to the semantic differences from illumination and subtle alignment differences in different phases.

To ensure the semantic continuity of the positions to be compared in the before and after period images as much as possible, we had to use the PDC structure to achieve the initial change position localization. The formula can be denoted as follows:

$$T_c = \left[ \mathbf{f}_{k \times k}\left(\mathbf{T}_1^1, \mathbf{T}_2^1\right); \mathbf{f}_{k \times k}\left(\mathbf{T}_1^2, \mathbf{T}_2^{12}\right); \cdots ; \mathbf{f}_{k \times k}\left(\mathbf{T}_1^N, \mathbf{T}_2^N\right) \right] \quad (1)$$

$$T_{pdc} = \mathbf{F}_{1 \times 1}\left(T_c\right) \quad (2)$$

where $\mathbf{f}_{k \times k}$ denotes group convolution using the convolution kernels of size k and $\mathbf{F}_{1 \times 1}$ denotes standard convolution with kernels of size 1. The number of convolution kernels was consistent with the number of channels $N$ of the original features $\mathbf{T}_1^1, \mathbf{T}_2^1$. First, the spatial fusion of features was performed channel by channel using a group convolution and then the correlation between the channels was calculated using a $1 \times 1$ convolution for the fused feature maps.

$$T_{1attn} = \mathbf{F}^{relu}_{3 \times 3}\left(T_{pdc} \otimes T_1\right) \quad (3)$$

$$T_{2attn} = \mathbf{F}^{relu}_{3 \times 3}\left(T_{pdc} \otimes T_2\right) \quad (4)$$

After the initial localization of the change position was completed, the fused feature map was multiplied by the previous feature maps and subjected to a $3 \times 3$ convolution to calculate the spatiotemporal attention. The details of the depth-separable change feature extraction are given in Figure 2a,b, which show the implementation details of the overall attention mechanism.

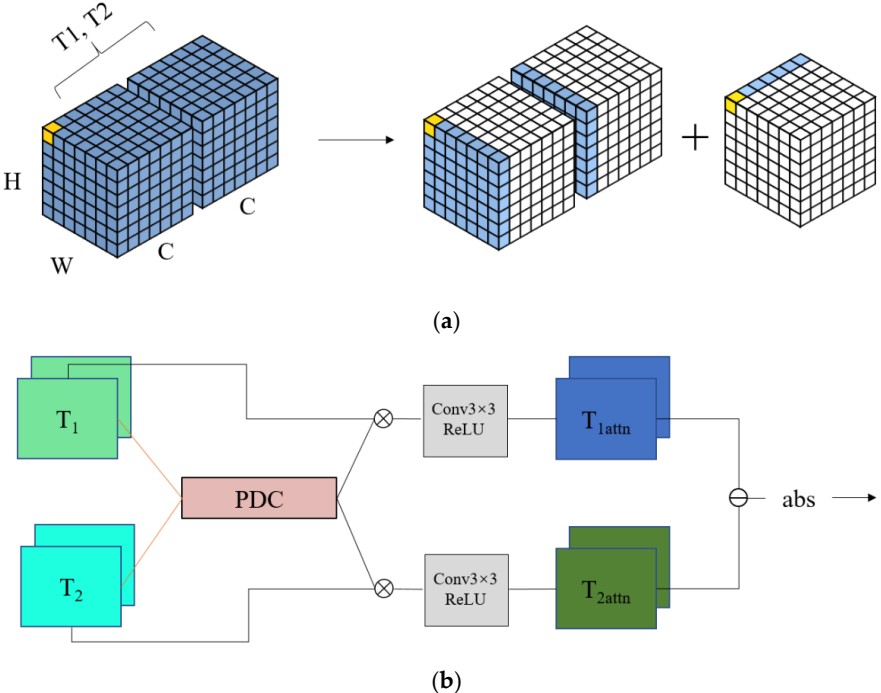

(a)

(b)

**Figure 2.** Decoder structure in the CD model. (**a**) PDC: a $7 \times 7$ convolution kernel size implementation of the PDC effect. The blue part represents the range in the space as well as the channel relationship considered to find a point (yellow point). The blue color on the left represents the computation of spatial position relations and that on the right represents the computation between features in different channels. (**b**) Details of the PDA decoder.

Throughout the PDA decoder, we did not introduce BN or LN to normalize the features; our proposed PDC had to introduce only a small number of model parameters to increase the feature representation of the model in space and channel locations compared to directly finding the absolute value of features or the Euclidean distance between features. We validated this module with the features in the PDA decoder (Figure 3). The visualization of the feature maps showed that PDA did prevent the introduction of redundant variation noise and computed more accurate variation features.

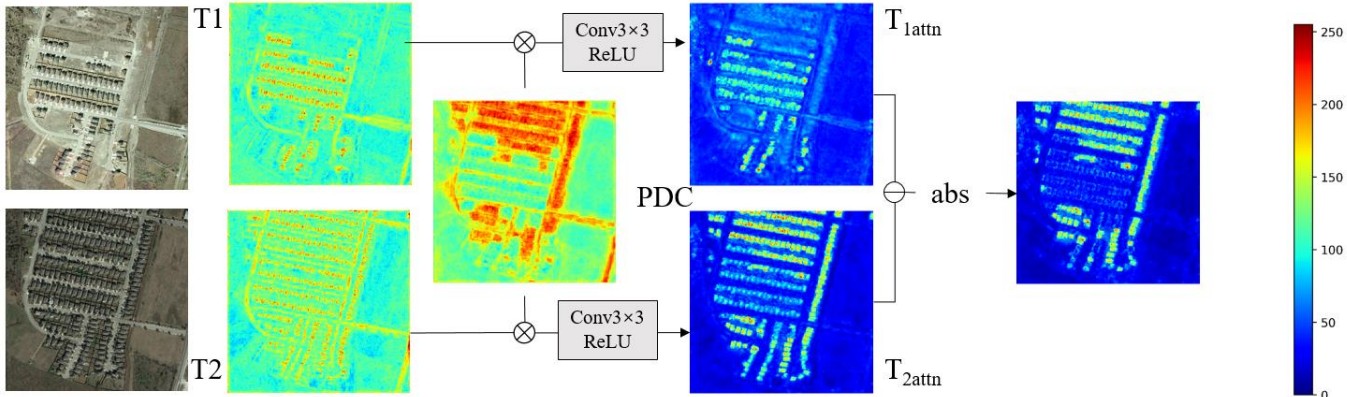

**Figure 3.** Visualization of the features in the PDA decoder. $T_1$ and $T_2$ denote the before- and after-period feature maps of the input. The PDC in the third column is the change attention feature map generated under a depth-separable difference pre-extraction. $T_{1attn}$ and $T_{2attn}$ in the fourth column represent the feature maps generated after the attention calculation. The plots in the last column indicate the final generated change feature maps.

### 3.4. Classification Head

The role of the classification header section was to perform pixel-by-pixel classification of the fused feature maps. This process is often accompanied by two operations: reducing the number of channels and resampling back to the original remote sensing image resolution. The former is used to output the final change probability map and the latter ensures the loss calculation in the pixel-by-pixel classification process. The structure is shown in Figure 1; we used a residual join operation after two layers of convolution of the fused features to increase the robustness of the model. Finally, the feature map was sampled to the original image size using bi-quadratic linear interpolation; a $1 \times 1$ convolution was used to complete the final classification.

### 3.5. Loss Function

Since the CD process used in this paper was considered a semantic segmentation task, the dataset in this paper suffered from some class imbalance. Here, we used the sum of the softmax cross-entropy loss and Dice loss as the final loss function. By default, this combination was used in all experiments in this paper.

#### 3.5.1. Dice Loss

The Dice loss was named after the Dice coefficient [46], which is a measure function used to assess the similarity of two samples; a larger value means that the two samples are more similar. The Dice loss function is formulated as follows:

$$L_{\text{Dice}} = 1 - \frac{2\sum_{i=1}^{N} x_i \hat{y}_i}{\sum_{i=1}^{N} x_i + \sum_{i=1}^{N} \hat{y}_i} \tag{5}$$

where $x_i$ and $\hat{y}_i$ denote the label value and predicted value of pixel $i$, respectively; and $N$ is the total number of pixels, which is equal to the number of pixels in a single image multiplied by the batch size.

### 3.5.2. Cross-Entropy Loss

As the CD process used in this paper was treated as a multiclass semantic segmentation task, we used cross-entropy loss in the training phase for implementation. The loss formula is as follows:

$$l_{ce} = \frac{1}{N} \sum_{i=1}^{N} [y_i \times \log x_i + (1 - y_i) \times \log(1 - x_i)] \tag{6}$$

where, similar to the Dice loss, $y$ is the true value of a point (usually 0 or 1), $x$ is the predicted value of a point, $N$ is the batch size multiplied by the total number of pixels on a single image, and $w$ is the weight given to each batch.

## 4. Experiments

We validated the model on two publicly available building CD datasets: the LEVIR-CD [17] dataset and the SYSU-CD [20] dataset. The experimental results showed that the proposed model outperformed other recently proposed CD methods.

In this section, we begin by introducing the experimental dataset. Then, we describe the details of our implementation and present the utilized evaluation metrics. Finally, we compare our method with some other methods. For ease of presentation, we present here an overview of some of the abbreviations used in the following sections. "XX-CD" indicates the XX experimental dataset. The proposed network was named PDACN based on the general structure of the network; this network with the three different encoder structures (ResNet18-S3, ResNet18-S4, and SegFormer-b0) was respectively named PDACN-R18S3, PDACN-R18S4, and PDACN-Segb0. Regarding the PDACN that appears in later figures, we defaulted to that with the encoder structure that yielded the best detection accuracy.

### 4.1. Dataset Settings

#### 4.1.1. LEVIR-CD Dataset

LEVIR-CD is an open dataset containing 637 ultrahigh-resolution (0.5 m resolution) Google Earth image pairs with $1024 \times 1024$ pixels. Images of 20 different locations in several cities in Texas were collected from 2002 to 2018; the image pairs ranged from 5 to 14 years. Architecture-related changes include building development (changes from soil/grassland/hardened ground or areas under construction/new building areas) and building decay. The dataset covers various types of buildings such as villas, high-rise apartments, small garages, and large warehouses. The dataset contains a total of 31,333 individual building changes with an average of approximately 50 building changes per image pair and an average size of approximately 987 pixels per change area. Note that most of the changes were due to building growth. The author of LEVIR-CD provided a standard training/validation/test split that assigned 70% of the samples for training, 10% for validation, and 20% for testing.

#### 4.1.2. SYSU-CD Dataset

This dataset contains 20,000 pairs of aerial images that are $256 \times 256$ in size and 0.5 m in resolution that were taken in Hong Kong between 2007 and 2014. The main types of changes in SYSU-CD include: (a) newly built urban buildings; (b) suburban dilation; (c) groundwork before construction; (d) vegetation changes; (e) road expansion; and (f) sea construction. Therefore, in addition to urban expansion and renewal-related change types, this dataset introduced some natural change types, which further increased the difficulty of CD. The entire dataset was divided at a ratio of 6:2:2; we ultimately obtained 12,000 training images, 4000 validation images, and 4000 test images. Table 2 shows the basic information for these two datasets.

### 4.2. Training Details

Our model was PyTorch-based and trained with mixed precision in the Ubuntu 20.04 OS using one NVIDIA Tesla V100 GPU. For the encoder consisting of ResNet [13], we used the official pre-training weights published by torchvision. For the encoder composed

by Segformer [45], we used pre-training weights from training on the Citispace dataset, in which the images are sized at 1024 × 1024. The other modules in the network structure were randomly initialized. During training, we performed random horizontal and vertical flips and on-the-fly rotations (90° | 180° | 270°) of the images with a probability of 0.5. By default, the batch sizes of images sized at 1024 and 256 were 2 and 16, respectively, during the training process. The model was fully trained at 300 epochs using the AdamW optimizer with a weight decay of 0.01. During this period, the learning rate decreased linearly from $3 \times 10^{-2}$ with the epoch to 0. At the end of each epoch, we performed a metric evaluation on the validation set and saved the highest weight of the relevant metric as the final result. In addition, we terminated the training if the highest value of the validation set metric still had not been achieved after 50 epochs.

**Table 2.** A brief introduction to the two datasets.

| Name | Bands | Image Pairs | Resolution (m) | Image Size | Train/Val./Test Set |
|---|---|---|---|---|---|
| SYSU-CD | 3 | 20,000 | 0.3 | 256 × 256 | 12,000/4000/4000 |
| LEVIR-CD | 3 | 637 | 0.5 | 1024 × 1024 | 445/64/128 |

*4.3. Evaluation Metrics*

To compare the performance of our model with the performances of other methods, we report their F1 and IoU scores with regard to the change class as the primary quantitative indices. Additionally, we report the precision and recall values for the change category of the CD task. The IoU and F1 values ranged from 0 to 1; the higher each value was, the better the performance. The IoU and F1 scores were calculated as follows, where TP denotes true positives, FP denotes false positives, and FN denotes false negatives:

$$IoU = \frac{TP}{TP + FP + FN} \tag{7}$$

$$F1 = 2 * \frac{\text{precision} * \text{recall}}{\text{precision} + \text{recall}} \tag{8}$$

The precision was calculated as:

$$\text{precision} = \frac{TP}{TP + FP} \tag{9}$$

The recall was calculated as:

$$\text{recall} = \frac{TP}{TP + FN} \tag{10}$$

We plotted the performances of PDACN-R18S3, PDACN-R18S4, and PDACN-Segb0 on the validation sets as shown in Figure 4. Since the sizes of the two datasets were very different, we chose the step size as the horizontal coordinate here for better presentation. As can be seen in Figure 4, PDACN achieved the best performance below 20,000. To further validate our model on the test set, we used the highest weight of the validated F1 as the checkpoint for the test.

*4.4. Results*

4.4.1. Comparison with Other Methods on LEVIR-CD

In this section, we present a comparison of the results of PDACN and other CD methods for the application to LEVIR-CD. FC-EF and its two variants (FC-Siam-Conc and FC-Siam-Diff) were not evaluated in the original paper on LEVIR-CD for the metrics. To ensure a consistent comparison of these methods, we fully retrained the methods used for comparison using the training approach and loss functions in this paper; both were validated on the same test set.

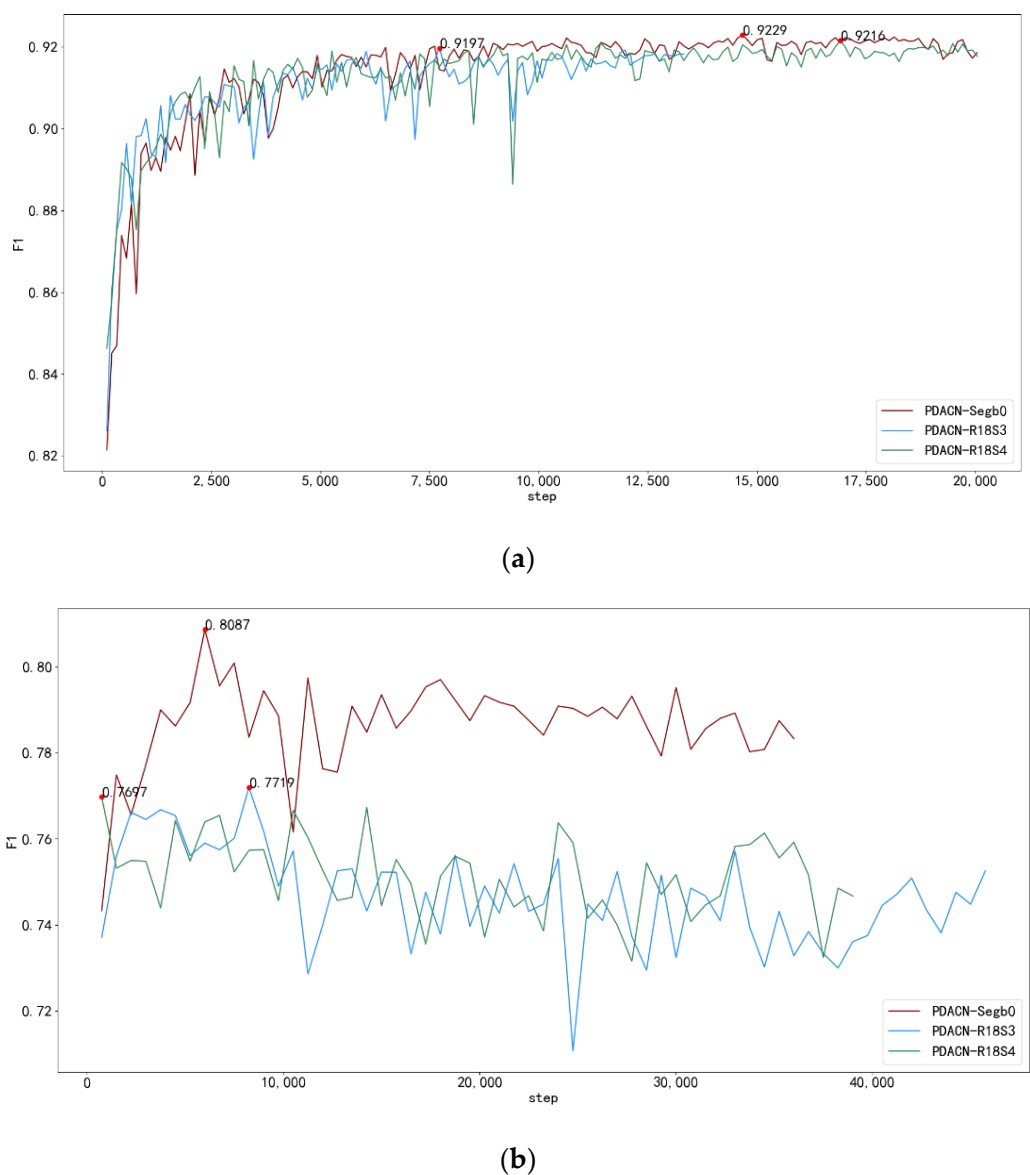

**Figure 4.** The performance of PDACN on the validation sets: (**a**) validation on the LEVIR-CD dataset; (**b**) validation on the SYSU-CD dataset.

Here, the quantitative evaluation results are given in terms of F1, IoU, precision, and recall, which are commonly used in CD task metrics; the results are shown in Table 3. Note that we used two inference testing strategies for image size (LEVIR-CD$_{256}$ and LEVIR-CD$_{1024}$) in the test set. The first one ensured that the test environment used in the original paper [38] method was available; i.e., the test set images were cropped into 2048 non-overlapping 256-sized image blocks for testing. The second one directly used 128 1024-sized test images for testing.

By using the LEVIR-CD$_{256}$ approach to test the model performance, the BIT approach achieved slight improvements of 0.81% and 1.33% over the F1 and IoU metrics of the original paper, respectively, which we believe was probably due to the training hyperparameters such as the learning rate and training time used in this paper. Compared with LEVIR-CD$_{256}$, LEVIR-CD$_{1024}$ achieved a greater improvement in all metrics to different degrees, which we believe occurred because the second test image used a 1024-sized image from the original test set, which made the input of the network richer in global and texture information, thus improving the network's expressive power.

**Table 3.** Quantitative results of different methods on the LEVIR-CD test set.

| Method | LEVIR-CD$_{1024}$ | | | | LEVIR-CD$_{256}$ | | | |
|---|---|---|---|---|---|---|---|---|
| | F1 (%) | Precision (%) | Recall (%) | IoU (%) | F1 (%) | Precision (%) | Recall (%) | IoU (%) |
| FC-EF [35] | 87.61 | 88.84 | 86.40 | 77.94 | 86.87 | 88.48 | 85.32 | 76.79 |
| FC-Siam-Conc [35] | 87.96 | 89.65 | 86.34 | 78.51 | 87.17 | 89.05 | 85.37 | 77.26 |
| FC-Siam-Diff [35] | 89.13 | 90.57 | 87.73 | 80.39 | 88.34 | 90.16 | 86.59 | 79.11 |
| BIT [38] | 90.72 | 93.11 | 88.45 | 83.01 | 90.12 | 88.16 | **92.15** | 82.01 |
| PDACN-R18S4 | 92.13 | 92.87 | 91.40 | 85.41 | 91.81 | **93.25** | 90.41 | 84.86 |
| PDACN-R18S3 | 91.59 | 90.64 | **92.55** | 84.48 | 91.18 | 90.00 | 92.38 | 83.78 |
| PDACN-Segb0 | **92.27** | **93.52** | 91.06 | **85.65** | **91.85** | 93.09 | 90.63 | **84.92** |

The bolded values are the maximum values of the corresponding columns.

Here, we focus on analyzing the results of the 1024-size inference test, which are shown in Table 3 corresponding to LEVIR-CD$_{1024}$. FC-EF obtained the lowest F1 of 87.61% and IoU of 77.94%, while its variant in the form of a Siamese network structure (FC-Siam-Conc) obtained a slight performance improvement in IoU, which indicated that the Siamese network structure could improve the performance expression of the model using this dataset. When using the Siamese network in the case of taking the difference and absolute value, the FC-Siam-Diff network yielded a 1.52% and 2.45% higher F1 and IoU, respectively, compared to the FC-EF. We believe this was because in this building dataset, the semantic information of the buildings in the images from different periods was similar, so a simple absolute value difference was sufficient for change-information extraction. In contrast, concatenation introduced unnecessary noise to limit the model expression while increasing the number of model parameters. The BIT constructed spatiotemporal attention to enhance the features using a transformer before absolute-value differencing and obtained a 2.62% and 2.45% higher IoU and precision than those of FC-Siam-Diff. This showed that using feature enhancement before absolute value extraction could further reduce the noise difference between features in different time phases, thus improving the detection accuracy and reducing "pseudo-variation" caused by misclassification. Our method (PDACN) achieved the best results for all metrics of the methods tested. Compared with those of BIT, we achieved a 1.55%, 2.64%, and 2.61% higher F1, IoU, and recall, respectively. We believe that the BIT filtered some useful information during the feature-reinforcement process on this dataset, which resulted in some missing building identifications; while our PDACN reduced "pseudo-variation" by pre-locating the change locations while reinforcing the integrity of individual buildings in the change areas. Finally, it should be noted that our network exhibited the best performance with the encoder form SegFormer-b0, followed by ResNet18-S4 and ResNet18-S3.

Figure 5 provides a visual comparison of the various methods used on the LEVIR dataset, where red and green represent the false and missed detection parts, respectively. Since the encoder for SegFormer-b0 performed best, we visualized only PDACN-Segb0. From top to bottom, their names in the test set were test_20, test_45, test_21, test_77, test_ 80, test_103, and test_107. Siam-Conc was able to determine the obvious range of variation in general. FC-Siam-Conc improved the visual effect on building edges but was not robust to pseudo-variations caused by lighting and shadows. According to the first and second rows, our PDACN and the BIT yielded better building boundaries and complete building profiles compared to the FC series approach. We believe that this occurred because both were robust against the effects of lighting and alignment on the bi-temporal images due to building attention mechanisms. According to rows 3 and 4, PDACN yielded the fewest red and green parts of all methods, which indicated that our method maintained a good performance when detecting small buildings. According to rows 5, 6, and 7, all methods exhibited some lack of connectivity in the detection of relatively large buildings, which we believe was a misclassification phenomenon due to the lack of differentiation between bare soil and buildings.

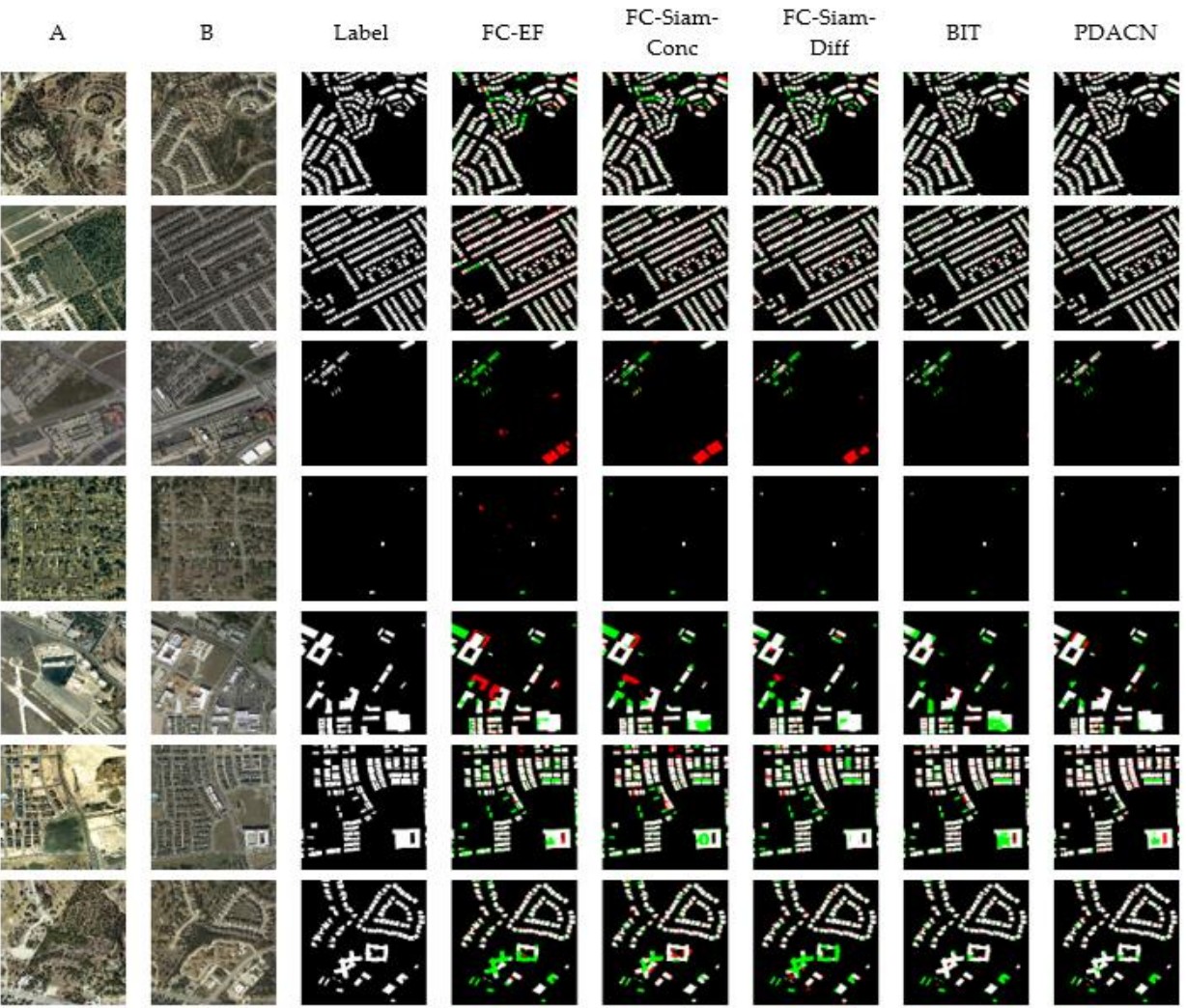

**Figure 5.** Performance of different methods on the LEVIR-CD dataset.

### 4.4.2. Comparison with Other Methods on SYSU-CD

In this section, we focus on comparison with DSAMNet [20], a method proposed together with SYSU-CD. Similar to the attention mechanism PDA proposed for the method in this paper, DSAMNet reinforces features by channel and spatial attention mechanisms before changing them. Here, we also used F1, IoU, precision, and recall as quantitative comparisons. We used the same training strategy as for LEVIR-CD except that the batch size was adjusted to 16. Since the data were at a fixed size of 256, the DSAMNet experimental results did not differ much from those of the original paper. Therefore, we directly cited the work related to the metrics of the model using the original paper. Among them, BiDateNet used LSTM to enhance temporal information between images and STANet included a spatiotemporal attention mechanism to enhance the features using ResNet18 as the encoder. DSAMNet, which also used ResNet18 as an encoder, additionally included shallow-feature supervised reinforcement through the introduction of spatial and channel attention mechanisms. These networks were essentially the same as the approach used in this paper, with the decoding part of the network modified on the basis of generating a change feature map.

As we can see in Table 4, compared to the LEVIR-CD single-building CD dataset, the F1, precision, recall, and IoU of our model (PDACN-Segb0) on the SYSU-CD dataset decreased by 10.27%, 11.62%, 8.96%, and 16.16%, respectively. This was because this dataset, in addition to significant artificial structures such as buildings alone, contained more intensive complex changes such as the expansion of built-up areas and bare soil

conversion of grass. This made the types of changes in complex scenes very expansive, which made the CD more difficult.

**Table 4.** Quantitative results of different methods on the SYSU-CD test set.

| Method | F1 (%) | Precision (%) | Recall (%) | IoU (%) |
|---|---|---|---|---|
| BiDateNet [36] | 76.94 | 81.84 | 72.60 | 62.52 |
| STANet [17] | 77.37 | 70.76 | 85.33 | 63.09 |
| DSAMNet [20] | 78.18 | 74.81 | 81.86 | 64.18 |
| PDACN-R18S4 | 79.09 | 83.08 | 75.46 | 65.40 |
| PDACN-R18S3 | 80.20 | 81.76 | 78.69 | 66.94 |
| PDACN-Segb0 | 82.55 | 84.63 | 80.57 | 70.28 |

In addition, our method achieved a better performance than DSAMNet in all metrics, especially in the case of the SegFormer-b0 encoder. The F1 and IoU were 6.10% and 4.37% higher, respectively, which showed that our method could still maintain excellent results in complex change scenes.

When we used the same encoder as DSAMNet (ResNet18), we obtained 0.91%, 8.27%, and 1.22% for F1, precision, and IoU, respectively, with the PDACN-R18S4 structure; and 2.02%, 6.95%, and 2.76% for F1, precision, and IoU, respectively, with the PDACN-R18S3 structure, respectively. Furthermore, the accuracy of the transformer-based encoder was better than that of the convolutional architecture and R18S3 with fewer network parameters returned better results than R18S4, which we believe was because the transformer-based network generalized better while the convolutional architecture was more likely to overfit this dataset.

Figure 6 shows a visual comparison between PDACN and DSAMNet on the SYSU-CD test dataset, where red and green represent false detections and missed detections, respectively. Here, we selected some representative types of changes in the test set for visualization (from top to bottom): vegetation changes (01536), pre-construction foundation work (01524), marine construction (00028), suburban expansion (00101), new urban buildings (00503), and road expansion (00147), followed by their names. Overall, compared with DSAMNet, the red parts in the last three columns of the figure show that our method yielded significantly fewer false detections under the transformer architecture. In particular, in the vegetation CD in row 1, DSAMNet incorrectly detected roads with a color mismatch caused by lighting factors while our PDACN was able to overcome this difficulty. The other types of CD, R18S3 and R18S4 with convolutional structures, still had false-detection performances similar to that of DSAMNet; while Segb0 with a transformer architecture performed the best, which indicated that the latter had a better generalization ability for complex change scenes.

*4.5. Ablation Study*

In this subsection, we report the results of an ablation study on the LEVIR-CD and SYSU-CD datasets regarding the PDA module to assess the effectiveness of each component presented in PDACN.

Based on the generation mechanism of the change feature map, our proposed PDA module utilized the simple absolute value difference (ABS), a simple $3 \times 3$ convolutional structure (CONV), and a PDC structure built based on depth-separable convolution. In the following experiments, "ABS" indicates that the network model performed only a simple absolute value operation on the feature map, "CONV" indicates that the model used a $3 \times 3$ convolution to simply reinforce the features, and "PDC" indicates that the model used the method of the pre-generation of significant change regions proposed in this paper to construct attentional mechanisms to strengthen the features.

As we can see in Table 5, the use of feature enhancement prior to ABS enhanced the CD performance of the model with a small increase in model parameters. In more detail, after adding the PDA module before utilizing ABS (0.154 M), the model F1 improved by

0.72% and the IoU by 1.23% on the LEVIR dataset and the F1 by 1.92% and the IoU by 2.71% on the SYSU-CD dataset, which indicated that using PDA before worsening the feature map could increase the model's fitting ability. In addition, adding the convolution operation after using attention to the PDA could further enhance the detection accuracy; we believe that this step could further eliminate the introduction of noise errors. Notably, the PDA module proposed in this paper assumed a significant role in the overall model performance improvement and facilitated the generation of more discriminative variance difference information.

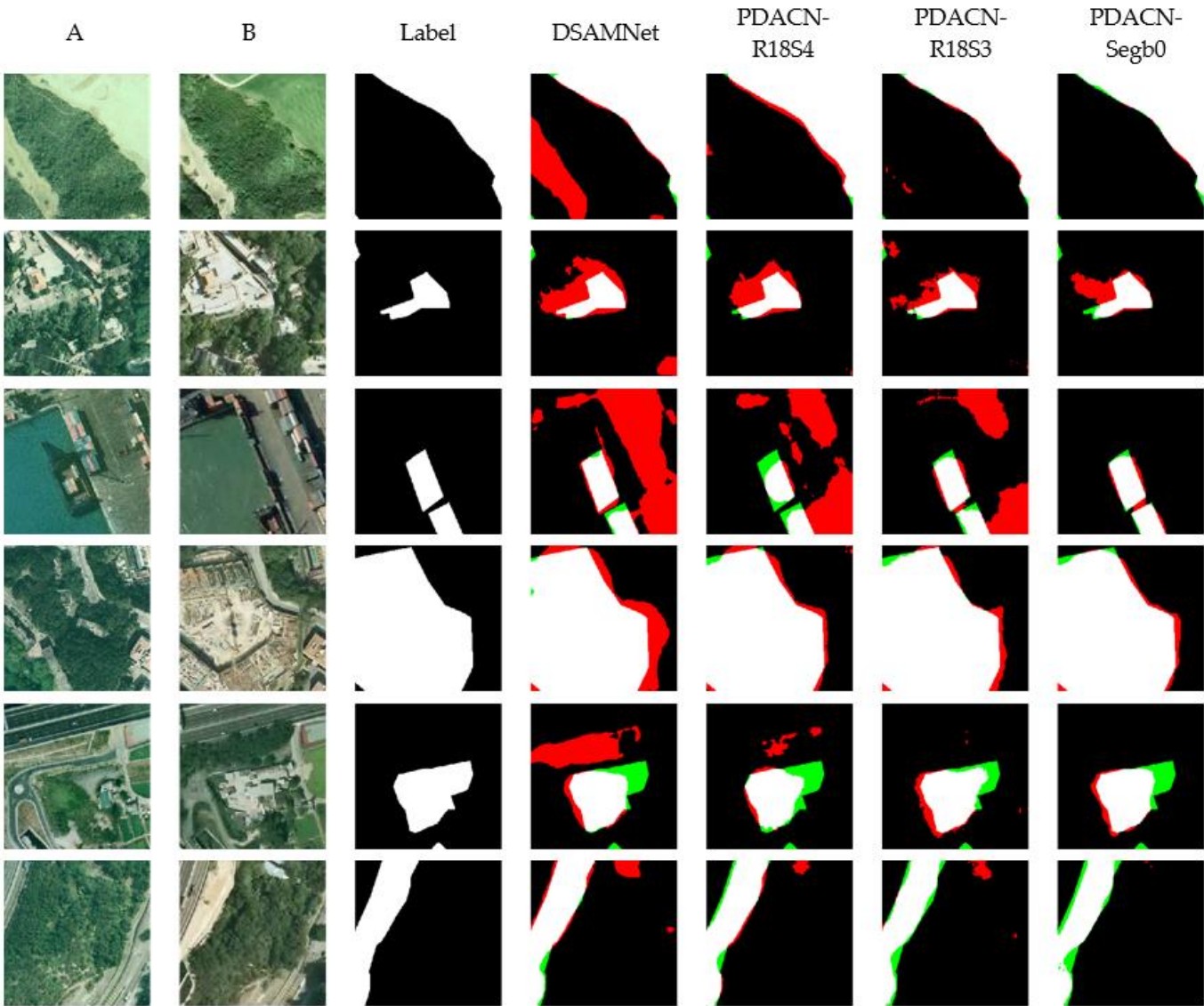

**Figure 6.** Performance of different methods on the SYSU-CD dataset.

**Table 5.** A study on the ablation of the PDA module in DACN-Segb0 networks. Experiments were conducted on the LEVIR-CD dataset and the SYSU-CD dataset.

| Method | Param (M) | LEVIR-CD | | SYSU-CD | |
|---|---|---|---|---|---|
| | | F1 (%) | IoU (%) | F1 (%) | IoU (%) |
| ABS | 0 | 91.31 | 84.01 | 80.01 | 66.68 |
| CONV + ABS | 0.017 | 91.89 | 85.04 | 80.85 | 67.86 |
| PDC + ABS | 0.154 | 92.03 | 85.24 | 81.93 | 69.39 |
| PDC+ CONV + ABS | 0.171 | 92.27 | 85.65 | 82.55 | 70.28 |

## 5. Discussion

### 5.1. Sensitivity Analysis of Convolution Kernel Parameters

To enable semantic continuity of the positions to be compared in the before and after images, we constructed a PDC structure for initial change position localization throughout the PDA attention mechanism. This structure first performed spatial feature fusion using convolution kernels on a channel-by-channel basis before calculating interchannel correlation using a $1 \times 1$ convolution for the fused feature maps. Therefore, the spatial feature fusion was sensitive to the choice of convolution kernel size here. To explore the effect of different convolution kernel sizes on PDACN, we conducted comparison experiments on two datasets by setting different values for the kernel size. The results are shown in Table 6.

**Table 6.** The effect of the convolution kernel size on the model structure in PDA. Experiments were conducted on the LEVIR-CD$_{1024}$ with a default image input size of 1024 and a DACN-Segb0 model.

| Kernel Size | LEVIR-CD | | SYSU-CD | |
| --- | --- | --- | --- | --- |
| | F1 (%) | IoU (%) | F1 (%) | IoU (%) |
| 1 | 92.05 | 85.26 | 81.93 | 69.39 |
| 3 | 92.12 | 85.38 | 81.90 | 69.47 |
| 5 | 92.27 | 85.65 | 82.55 | 70.28 |
| 7 | 92.09 | 85.34 | 81.52 | 68.81 |
| 9 | 91.87 | 84.96 | 81.71 | 69.07 |

### 5.2. Efficiency Test

Table 7 reports the number of parameters (Params.), floating point operations per second (FLOPS), and the F1/IoU scores for the different methods used on the LEVIR-CD test set. Our PDACN obtained the highest F1/IoU scores with the SegFormer-b0 encoder and maintained the characteristics of the lightweight model.

**Table 7.** Comparison of the computational effort of the models used in this paper. Experiments were performed on the LEVIR-CD256 dataset with default input images sized at $256 \times 256$.

| Method | Params. (Mb) | FLOPS (G) | F1 (%) | IoU (%) |
| --- | --- | --- | --- | --- |
| FC-EF | 1.35 | 3.55 | 86.87 | 76.79 |
| FC-Siam-Conc | 1.54 | 5.3 | 87.17 | 77.26 |
| FC-Siam-Diff | 1.35 | 4.7 | 88.34 | 79.11 |
| BIT | 3.55 | 10.60 | 90.12 | 82.01 |
| DSAMNet | 16.95 | 72.18 | - | - |
| DACN-R18S4 | 12.13 | 36.78 | 91.81 | 84.86 |
| DACN-R18S3 | 3.65 | 31.91 | 91.18 | 83.78 |
| DACN-Segb0 | 4.22 | 5.58 | 91.85 | 84.92 |

### 5.3. Future Work

The proposed method could accurately identify the change regions to be detected; the experimental results showed that our proposed method achieved a 6.1% and 4.4% improvement in the intersection over union (IoU) and F1 metrics, respectively, over those of the state-of-the-art method on the SYSU-CD public remote sensing image dataset. However, compared to those for the LEVIR-CD single-building CD dataset, the F1, precision, recall, and IoU of our PDACN-Segb0 model for the SYSU-CD dataset decreased by 10.27%, 11.62%, 8.96%, and 16.16%, respectively. This indicated that there was also a significant challenge in terms of the performance of this model using cross-domain datasets. In the future, we will develop more effective CD algorithms to improve the generalization ability of the model. In addition, the PDACN in this paper was studied under only two lightweight encoder architectures. In contrast, the transformer architecture had more obvious advantages. Next, we will continue to investigate the effect of boosting the network encoder in the model.

## 6. Conclusions

In this paper, we presented PDACN, an effective method for dual-time remote sensing CD. PDACN consisted of an encoder that could extract multiscale features, a decoder that focused on the semantic alignment between temporal features, and a classification head. In the decoder, we constructed a new convolutional attention structure, PDA, which was able to reduce the network's attention to unchanged regions before generating change features, thus reducing pseudo-change in the data source potentially caused by semantic differences due to illumination and subtle alignment differences. To demonstrate the effectiveness of the PDA attention structure, we designed lightweight network structures for encoders under both convolution-based and transformer architectures. The experiments were conducted on a single-building CD dataset (LEVIR-CD) and a more complex multivariate-change-type dataset (SYSU-CD). The results showed that our PDA attention structure resulted in the generation of more differentiated change variance information and that PDACN achieved the best performance results with the same level of network model parameters in the transformer architecture.

**Author Contributions:** Conceptualization, B.L. and G.W.; methodology, B.L.; validation, G.W. and T.Z.; writing—original draft preparation, B.L. and G.W.; writing—review and editing, B.L. and T.Z.; supervision, H.Y. and S.Z. All authors have read and agreed to the published version of the manuscript.

**Funding:** This work was funded by the Application Demonstration System of GaoFen Remote Sensing Mapping of China (No. 42-Y30B04-9001-19/21) and the Science and Technology Talent Project of the Ministry of Natural Resources of China (No. 121106000000180039-2206).

**Institutional Review Board Statement:** Not applicable.

**Informed Consent Statement:** Informed consent was obtained from all subjects involved in the study.

**Data Availability Statement:** Data associated with this research are available online. The LEVIR-CD dataset is available for download at https://justchenhao.github.io/LEVIR (accessed on 4 October 2022). The SYSU-CD dataset is available for download at https://github.com/liumency/DSAMNet (accessed on 4 October 2022).

**Conflicts of Interest:** The authors declare no conflict of interest.

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
