# Peer review of "Remote Sensing Image-Change Detection with Pre-Generation of Depthwise-Separable Change-Salient Maps"

_remotesensing, doi:10.3390/rs14194972_

Round 1

Reviewer 1 Report

The text is relevant to the journal. However, I have some suggestions to improve the readers’ understanding of the text.

 1 - The introduction ends with the contributions; I suggest ending with the objectives. Contributions could be included in the discussion highlighting the quantitative results.

 2 –Related work – I suggest including an opening paragraph describing the different approaches to change detection that will be described in the subtopics. For example, spatial unit (pixel or object), input data (direct concatenation of temporal bands or change intensity information (CVA, change measures or differencing by the log-ratio operator)), and methods for extracting features. I believe topic 2.3 may precede 2.2 for better reader understanding.

 3 – Perhaps a previous explanation explaining the General AI-Based Change Detection Frameworks can help to understand the article (see Shi et al 2020).

 4 – Figures present several acronyms but do not explain them in the legend.

 5 – The acronyms PDC, PDA, and PDACN have different meanings in the text. Perhaps Figure 1 could illustrate what the PDA and PDACN include.

 6 - Convolutional structure (COV) or (CONV) (Line 509)

 7 – Authors should better explain Absolutely (ABS). It is related to the simple absolute value difference.

 8 – The discussion text does not connect to other studies. What does the present study advance compare to other studies? I believe it should include limitations of the study and future work.

Minor corrections

 Line 12 - “In this paper, we propose an effective remote CD method for dual-time remote sensing CD. It consists of an encoder…” I suggest “This research proposes an effective bi-temporal remote sensing CD comprising an encoder...”

 Line 15 “PDA” put the meaning of the acronym.

 Line 22 “The results show...structure results in the generation of” I suggest “The results show…structure generates”

 Line 32 - “The definition of change in this process” I believe the phrase does not refer to a definition. I suggest “Change analysis can be restricted to a target, such as building [2], mudflow and landslide [3], building damage assessment [4,5], land cover [6], and deforestation [de Bem et a. 2020; Torres et al., 2021].” or ”Typically, change analysis is restricted to one target, such as construction building [2], mudflow and landslide [3], building damage assessment [4,5], land cover [6], and deforestation [de Bem et a. 2020; Torres et al., 2021].”

An interesting point would be to include the topic of deforestation that has high interest (de Bem et a. 2020; Torres et al., 2021).

de Bem, P.P.; de Carvalho Junior, O.A.; Fontes Guimarães, R.; Trancoso Gomes, R.A. Change Detection of Deforestation in the Brazilian Amazon Using Landsat Data and Convolutional Neural Networks. Remote Sens. 2020, 12, 901. https://doi.org/10.3390/rs12060901

Torres, D.L.; Turnes, J.N.; Soto Vega, P.J.; Feitosa, R.Q.; Silva, D.E.; Marcato Junior, J.; Almeida, C. Deforestation Detection with Fully Convolutional Networks in the Amazon Forest from Landsat-8 and Sentinel-2 Images. Remote Sens. 2021, 13, 5084. https://doi.org/10.3390/rs13245084

 Perhaps a description of binary and multiple CDs would be adequate (Saha et al., 2021).

Saha, S., Kondmann, L., Song, Q., & Zhu, X. X. (2021). Change detection in hyperdimensional images using untrained models. IEEE Journal of Selected Topics in Applied Earth Observations and Remote Sensing, 14, 11029-11041.

Line 35 - “With the development of satellite imaging technology, it has become increasingly easier to collect…” I suggest “The development of satellite imagery technology has facilitated the collection of…”

 Line 37 - “spectral spatial and temporal resolution” I suggest “spectral, spatial, and temporal resolution”

 Line 38 - “resolution along with the addition… makes CD” I suggest “resolution and the addition… make the CD…”

 Line 40 - “to reduce the interference” I suggest “to reduce interference”

 Line 41 - “Early traditional CD…” This sentence needs bibliographic references, preferably review articles.

 Line 43 - “some of them performed better” some? Missing bibliographic references.

 Line 51 - “dual-temporal” The most used term is “bi-temporal”. I suggest reviewing the entire text.

 Line 73-74 - “lightweight multiscale feature fusion change feature generation encoder structure” difficult to understand.

 Line 86-87 - “improvement on the” I suggest “improvement in the”

 Line 90 - “2.1. Bitemporal change detection in deep learning” Firstly, I suggest “2.1. Bitemporal change detection using deep learning “. Besides, the topic title could report the main issue addressed “networks based on the block structure x network structures for pixel-by-pixel classification.”

 Line 91-93 - Many or all? This sentence does not help.

 Line 93 – “over different time periods” redundancy “over different periods”

 Line 94-95 – Missing bibliographic references. Would the terms correlate with “pixel-level or object-level change detection methods” (Jiang et al. 2022)? (Per-pixel vs. object-based classification)

Jiang, H., Peng, M., Zhong, Y., Xie, H., Hao, Z., Lin, J., ... & Hu, X. (2022). A Survey on Deep Learning-Based Change Detection from High-Resolution Remote Sensing Images. Remote Sensing, 14(7), 1552.

 Line 93-94 “Network change detectors constructed based on these datasets can be classified from the perspective of processing units:“ or “Network change detectors can be classified by spatial units:”. I believe that “spatial units” are more suitable than “processing units”.

 Lines 96-132 - The text of topic 2.1. can be improved.

 Line 133 - “2.1….” wrong numbering “2.2. …”

 Line 135, 146, 220, 256, 431, 17 “pseudovariation” or “pseudo-variation”

 Line 136 – “in some external factors” I suggest “in external factors”

 Line 141 - “Song et al.[33]” insert a space “Song et al. [33]”

 Line 149 – “Chen et al. [35]used” insert a space “Chen et al. [35]used”

 Line 150, 231, 232, 58 “pretrained network” hyphen “pre-trained network”

 Line 158 - “2.1. Designing …” wrong numbering “2.3. Designing …“

 Line 160 – “metric-based learning and classification-based learning” I suggest including references.

 Line 172 - “the improvements that can be achieved by these methods are” I suggest “the improvements that these methods can achieve are

 Line 256 - “There are two problems that need” I suggest “Two problems need”

 Line 261 - “a number of researchers” I suggest “some researchers”

 Line 283 - “to achieve the calculation of” I suggest “to calculate”

 Line 321 – “3.5.1. Dice loss” “3.5.2. Dice loss”

 Line 330 – “CD [14]dataset” I suggest “CD [14] dataset”

 Line 418 - “due to the fact that” I suggest “because”

 Line 449 - “all methods exhibits” I suggest “all methods exhibit”

 Line 509 – “change predetection (PDC)” I suggest “PDC”. The term has already been defined in Line 198.

 Line 206 – “PDA change-detection network (PDACN)”. The term has already been defined in line 71 as "pre-generation of depthwise separable change salient maps (PDACN)".

References - Some citations are not consistent with the journal's guidelines. 3 / 12 / 19 / 21 / 22 / 30 / 31 / 41. 

Reviewer 2 Report

This paper presents a deep learning method for remote sensing image change detection. The topic is relevant in the field, because it plays an important role in many remote sensing applications, such as land use management and facility management. The novelty lies in the new attention structure that reduces the network's attention on unchanged regions.

Regarding the methodology, the improvement is the attention block.

The experimental results and the conclusions are consistent with the main research question they presented at the beginning. Also, references are appropriate and up-to-date.

Tables and figures are fine as they are self-explained. I was only concerned about the representation, but I deemed that it would not be a big problem because the Production Team would elaborate the final manuscript and double-check with the authors.

Specific comments are as follows:

1. The major contribution is the new attention network but its description is quite obscured. Hence, it would be better to detail the PDC and add a more intuitive description of the PDA.

2. It is advisable to plot the precision-recall curve to gain more insights into experimental results.

3. Abbreviations should be defined on their first occurrence.

Reviewer 3 Report

This paper proposed a Siamese network for dual-time remote sensing change detection, which take advantage of a PDA module to pay more attention from change regions for minimize pseudo-variation. The method output the change saliency map based on the encoder-decoder-classification structure. A lightweight encoder module (SegFormer-b0) is used for feature extraction instead of the standard encoder ResNet18. In order to address the problem of bias between diachronic features, the PDA decoder is proposed as the main highlights of this paper. The PDA multiply the feature map with the spatial weights extracted by the PDC module to obtain the final feature map. Finally, the classification head output the change saliency map. Generally speaking, the treatment of potential pseudo-variation in this paper is interesting. The presented method has sufficient experiments and the manuscript is well structured. However, the following comments and remarks must be addressed:

1.     The PDA decoder is the main highlights of this paper, but the description of PDC module is not very detailed. Can you explain the calculation process in detail by formulas and graphs?

2.     The method mainly solves the two problems raised in Section2.3. However, there is no test and analysis for this problem in the designed experiment.

3.     The PDC module implements the attention mechanism with depth-wise separable convolution. Can you give a theoretical explanation? If you just want to learn weights why not use a module like self-attention?

4.     The number of samples in the two datasets is quite different. The LEVIR-CD datasets has larger size but less samples. Does this affect the final performance of the network?

5.     Not only the kernel size can affect the network performance, but also the number of kernels. Have you tested the optimal number of kernels?

Round 2

Reviewer 3 Report

After revision, I think the article meets the requirements of the journal.

Author Response

Thank you very much for your work in reviewing our paper.